# Position: Academic Conferences are Potentially Facing Denominator Gaming Caused by Fully Automated Scientific Agents

Rong Shan [* 1]   Te Gao [* 2]   Hang Zheng [1]   Yunjia Xi [1]   Jiachen Zhu [1]   Zeyu Zheng [3]   Yong Yu [1]   Weinan Zhang [1]   Jianghao Lin [1]

## Abstract

The implicit policy of maintaining relatively stable acceptance rates at top AI conferences, despite exponentially growing submissions, introduces a critical structural vulnerability. This position paper characterizes a new systemic threat we term *Agentic Denominator Gaming*, in which a malicious actor deploys AI agents to generate and submit a large volume of superficially plausible but low-quality papers. Crucially, their objective is not the acceptance of low-quality papers, but rather to inflate the submission denominator and overwhelm reviewing capacity. Under a relatively stable acceptance rate, this dilution can systematically increase the publication probability of a small, targeted set of legitimate papers. We analyze the practical feasibility of this threat and its broader consequences, including intensified reviewer burnout, degraded review quality, and the emergence of industrialized automated *agent-driven paper mills*. Finally, we propose and evaluate a range of mitigation strategies, and argue that durable protection will require system-level policy and incentive reforms, rather than relying primarily on technical detection alone.

## 1. Introduction

Premier academic conferences serve as core infrastructure for validating and disseminating scientific knowledge. In recent years, rapid advances in *scientific AI* have sharply lowered the cost of producing conference-style manuscripts at scale. Agentic systems for literature synthesis, experiment planning, and autonomous research workflows are becoming increasingly realistic (Ren et al., 2025; Yamada

[*]Equal contribution [1]Shanghai Jiao Tong University [2]Central South University [3]Carnegie Mellon University. Correspondence to: Jianghao Lin <linjianghao@sjtu.edu.cn>.

*Proceedings of the 43rd International Conference on Machine Learning*, Seoul, South Korea. PMLR 306, 2026. Copyright 2026 by the author(s).

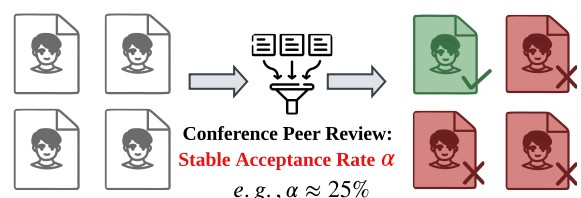

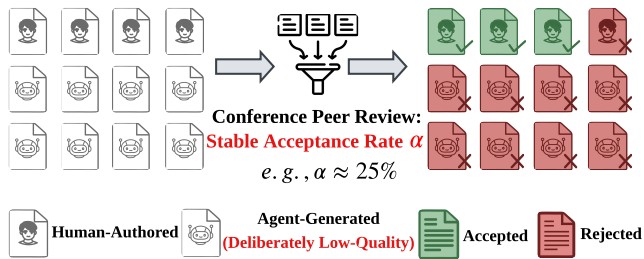

*Figure 1.* An illustration of the mechanism of **Agentic Conference Denominator Gaming**. Assume a conference maintains a stable acceptance rate of around 25%. If the denominator increases from 4 to 12, the number of accepted papers must rise from 1 to 3. Without an influx of new high-quality submissions, this mechanism forces the conference to lower its quality threshold, effectively *rescuing* human-authored papers that were originally at or even below the borderline.

et al., 2025; Huang et al., 2025). Furthermore, automated tools for paper writing, debugging, and refinement have reduced the marginal cost of paper iteration (Hao et al., 2026; Hou et al., 2025; Sontake, 2025; Silva et al., 2025). As a result, the traditionally labor-intensive craft of manuscript preparation is evolving into a streamlined, automated workflow, where Generative AI and autonomous agents enable rapid and large-scale production.

However, the shift can introduce a new and under-appreciated systemic risk, which we term **Agentic Conference Denominator Gaming**. Under the long-standing norm that elite venues maintain relatively stable, low acceptance rates as a signal of selectivity (Wang, 2025; Chen & Konstan, 2010), the total number of submissions becomes a

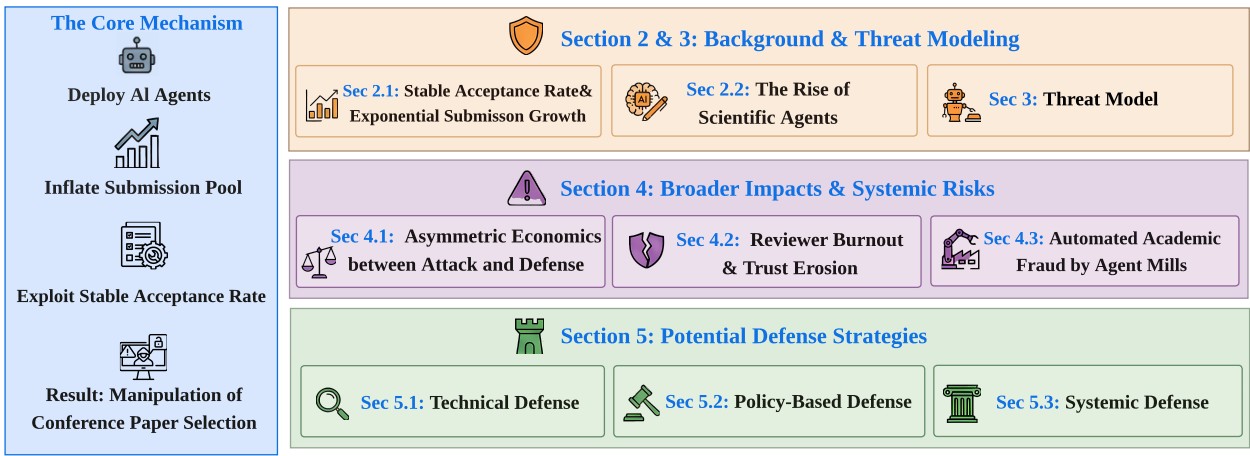

*Figure 2.* The structure of our position paper.

controllable *denominator* that shapes each paper's marginal acceptance probability. A malicious actor could exploit this norm by deploying AI agents to generate and submit large volumes of plausible-but-low-value manuscripts at negligible marginal cost. Crucially, these submissions are **not** intended to be accepted; instead, they function as strategic noise that inflates the submission pool and thereby improves the relative odds of a targeted set of human-authored papers (shown in Figure 1). The issue is not simply more AI-generated papers, but a transition from content inflation (Lin et al., 2025a) to *mechanism manipulation*. In other words, the same tools that accelerate the publication pipeline can be weaponized to erode the integrity of the pipeline itself, shifting the competitive selection not through better science, but through deliberate manipulation of the stable-acceptance-rate mechanism.

The implications extend far beyond the manipulation of selection outcomes at a single conference. First, the threat exploits a stark economic asymmetry: attackers can generate and submit large volumes of plausible manuscripts at low cost, while the academic community bears the defensive burden as thousands of hours of uncompensated expert labour. The imbalance can even scale across venues due to the reusability of rejected agentic submissions, creating a cascading ecosystem risk in the publication pipeline. Second, in a peer-review system already strained by reviewer fatigue (Chen et al., 2025; Adam, 2025; Petrescu & Krishen, 2022; Horta & Jung, 2024; Parrish, 2024; Kim et al., 2025), a flood of low-quality submissions would sharply degrade the signal-to-noise ratio, accelerate burnout, and drive experienced reviewers away, thereby eroding review quality and the reputational value of acceptance itself. Third, it represents an escalation from *human paper mills* to fully *automated agent-driven paper mills*, collapsing the cost of academic fraud while increasing its scale and evasiveness, and pushing the community toward an epistemic security

crisis where scientific authenticity can no longer be taken for granted.

Based on these discussions, we state our core position: **Academic conferences are potentially facing a new systemic threat:** *Conference Denominator Gaming* **enabled by fully automated scientific agents.** The purpose of this position paper is not to advocate for or enable adversarial behaviors. Instead, by formalizing the mechanism and objectives of *Agentic Conference Denominator Gaming*, we aim to surface an under-discussed vulnerability in the modern publication ecosystem and motivate proactive, system-level mitigations *before* such attacks become routine.

The remainder of this paper is organized as follows, which is shown in Figure 2. Section 2 reviews conference submission trends, acceptance-rate stability, and recent progress in automated scientific agents, based on which we introduce the mechanism of the threat. Section 3 formalizes the denominator-gaming threat model, which can be implemented as a multi-agent pipeline. Section 4 analyzes broader impacts such as reviewer overload, trust erosion, and economic asymmetry. Section 5 discusses potential defenses comprehensively. Section 6 addresses key alternative viewpoints. Finally, Section 7 concludes with recommendations for building a more robust academic publication infrastructure.

## 2. Agentic Conference Denominator Gaming

The feasibility of the Agentic Conference Denominator Gaming is predicated on two key trends: the relatively stable acceptance rate of modern conferences, and the rapid advancement of AI's scientific research capabilities. The threat mechanism emerges precisely under these trends.

## 2.1. Submission Growth and Stable Acceptance Rates

In the last decade, many top-tier conferences have been overwhelmed by an exponential submission growth (Joanne, 2025). NeurIPS, for example, saw its submission count swell from about 6,700 in 2019 to over 30,000 in 2025, representing a compound annual growth rate of approximately 29%. Despite the submission number surge, acceptance rates have remained remarkably stable. As illustrated in Figure 3, venues such as CVPR, NeurIPS, and EMNLP have consistently maintained acceptance rates in the 20-30% range for many years. We further dive into the detailed statistics and compute the standard deviations, which are shown in Table 1. The stability is visible among different AI scopes, and many conferences (*e.g.*, NeurIPS, ACL, and SIGIR) have a notably small standard deviation (*i.e.*, < 1).

This stability is not a statistical inevitability but an implicit policy choice (Butler & Spoelstra, 2020; Wang, 2025; Chen & Konstan, 2010). The stable acceptance rate is crucial for a conference's normal operation and sustainability of prestige. However, this policy creates a predictable and exploitable environment. Decision thresholds are often calibrated to align with historical norms, causing the acceptance rate to function less as a quality filter and more as an administrative target. This mechanism serves as the structural foundation for the threat described in this paper.

## 2.2. The Rise of Scientific Agents

The capability of AI to generate plausible scientific papers is no longer theoretical. Recent advancements in LLMs and agentic AI have led to the automation of substantial portions of the research lifecycle (Ren et al., 2025; Huang et al., 2025; Lee et al., 2024). Frameworks of *Scientific Agents* have demonstrated the ability to take a research idea and autonomously perform a literature review, conduct experiments, and write a full report (Ren et al., 2025; Ghafarollahi & Buehler, 2025; Reddy & Shojaee, 2025; Xi et al., 2025).

Beyond simple text generation, agents can be built to search academic databases like arXiv, summarize findings, and compile the results into a LaTeX PDF, even self-correcting compilation errors (Guo et al., 2026; Yang et al., 2025b; Zhou et al., 2026; Zhang et al., 2024). The *AI Scientist* framework, for instance, claims to automate the entire process from idea generation to paper writing and even peer review, at an estimated cost of only $15 per paper (Lu et al., 2024; Yamada et al., 2025). While specialized scientific LLMs like Galactica have faced challenges with accuracy, they demonstrated a powerful ability to reason about scientific knowledge, generate LaTeX equations, and process citations (Taylor et al., 2022; Wei et al., 2025). These tools and frameworks prove that generating thousands of stylistically plausible, correctly formatted, and topic-relevant papers is now technically and economically feasible.

*Table 1. Main track* acceptance rate statistics of AI conferences in different scopes. We report the mean, variance (*Var*), and standard deviation (*std*) from 2021 to 2025. *Std* less than 1 is marked as blue, indicating that the acceptance rate is remarkably stable.

| Category | Conference | Mean | Var | Std |
|---|---|---|---|---|
| Artificial Intelligence | AAAI | 20.62 | 10.23 | 3.20 |
| | IJCAI | 15.26 | 4.18 | 2.05 |
| Computer Vision & Pattern Recognition | CVPR | 24.09 | 1.71 | 1.31 |
| | ECCV | 24.06 | 0.11 | 0.33 |
| | ICCV | 25.55 | 1.29 | 1.13 |
| Data Mining & Information Retrieval | CIKM | 23.89 | 3.39 | 1.84 |
| | ICDM | 10.71 | 2.31 | 1.52 |
| | KDD | 18.34 | 6.90 | 2.63 |
| | RecSys | 18.67 | 2.92 | 1.71 |
| | SIGIR | 20.78 | 0.69 | 0.83 |
| | WWW | 19.73 | 1.05 | 1.03 |
| | WSDM | 17.45 | 0.82 | 0.90 |
| Machine Learning & Learning Theory | AISTATS | 29.47 | 1.44 | 1.20 |
| | COLT | 34.05 | 2.06 | 1.44 |
| | ICLR | 31.15 | 1.76 | 1.33 |
| | ICML | 25.17 | 8.09 | 2.84 |
| | NeurIPS | 25.53 | 0.28 | 0.53 |
| | UAI | 30.34 | 2.20 | 1.48 |
| Natural Language Processing | ACL | 21.15 | 0.36 | 0.60 |
| | EMNLP | 21.49 | 1.44 | 1.20 |
| | NAACL | 23.34 | 4.06 | 2.01 |
| Robotics & Automation | ICRA | 42.75 | 4.45 | 2.11 |
| | RSS | 17.42 | 2.54 | 1.59 |
| Speech & Signal Processing | ICASSP | 46.39 | 2.40 | 1.55 |
| | INTERSPEECH | 49.51 | 1.47 | 1.21 |

## 2.3. The Mechanism and Objective of Agentic Conference Denominator Gaming

The core mechanism of Agentic Conference Denominator Gaming is simple: the attacker uses a multi-agent pipeline to inflate the submission denominator by flooding the venue with large volumes of AI-generated manuscripts, which appear superficially plausible but are intentionally low-quality. These submissions act as disposable *cannon fodder*, which are not expected to be accepted but expand the total number of submissions.

Under a relatively stable acceptance rate, the number of accepted papers is tied to the total submission count. When the submission pool is artificially inflated, the conference is forced to increase the absolute number of acceptances (to preserve its historical acceptance rate). Since the injected papers are largely destined for rejection, they mainly occupy the rejection portion of the statistics. As a consequence, the acceptance boundary must move deeper into the remaining pool to fill the expanded acceptance quota, creating a statistical distortion in which the effective acceptance probability for the non-synthetic manuscripts is increased.

The **objective** of the attacker is therefore not to win acceptance for the mass-produced AI papers. Rather, the attacker seeks to improve the acceptance odds of a separate, smaller

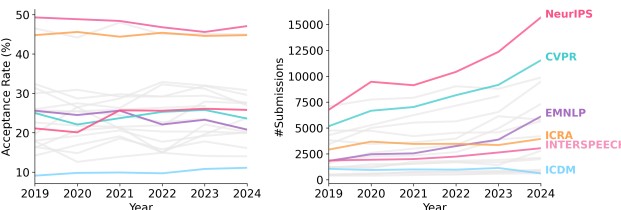

*Figure 3.* Acceptance rates and submission numbers of some AI conferences from 2019 to 2024. The displayed conferences include ACL, EMNLP, CVPR, ICCV, ICML, NeurIPS, ICLR, COLT, UAI, AISTATS, AAAI, IJCAI, KDD, SIGIR, WWW, INTERSPEECH, ICASSP and ICRA. Some of them are highlighted due to fast submission growth yet remarkably stable acceptance rate.

set of *targeted* manuscripts that they want published. From a game-theoretic perspective (Petrosyan & Yeung, 2019), academic publishing is a high-stakes, non-cooperative environment where publications function as career currency. The attacker behaves as a rational agent maximizing utility (accepted papers) by hacking the game parameters (*i.e.*, the stable acceptance rate), rather than improving intrinsic quality (Butler & Spoelstra, 2020; Faria & Goel, 2016).

**Mathematical formalization.** We provide a simple mathematical formalization to help clarify how denominator inflation affects the acceptance rate.

Let $N_h$ and $N_a$ denote the number of human and AI-generated submissions, respectively. The total number of submissions is $N_{sub} = N_h + N_a$.

Suppose the conference aims to maintain a stable target acceptance rate, denoted by $\alpha$. The total number of accepted papers $N_{acc}$ is determined by:

$$N_{acc} = \alpha \cdot (N_h + N_a). \tag{1}$$

The total accepted set consists of accepted human papers $N_{acc}^h$ and accepted AI papers $N_{acc}^a$:

$$N_{acc} = N_{acc}^h + N_{acc}^a. \tag{2}$$

Since the AI-generated papers are intentionally low-quality, we assume they are almost always rejected during the review process. Thus, $N_{acc}^a \approx 0$, and the quota for accepted papers is filled almost entirely by human submissions:

$$N_{acc}^h \approx N_{acc} = \alpha \cdot (N_h + N_a). \tag{3}$$

The effective acceptance rate for human submissions, denoted by $\hat{\alpha}$, then becomes:

$$\hat{\alpha} = \frac{N_{acc}^h}{N_h} \approx \frac{\alpha \cdot (N_h + N_a)}{N_h} = \alpha \cdot \left(1 + \frac{N_a}{N_h}\right). \tag{4}$$

As the ratio of AI submissions to human submissions $\frac{N_a}{N_h}$ increases, the effective human acceptance rate $\hat{\alpha}$ rises proportionally, even if the actual quality of human papers remains unchanged.

## 3. Threat Modeling

To demonstrate the practicality and fully understand the threat, we present the architecture of a proof-of-concept system engineered for automation and scalability. The design comprises two specialized agents that decouple the task of content creation from submission logistics.

- **Research Agent:** Responsible for the high-volume generation of distinct, stylistically plausible academic manuscripts.

- **Submission Agent:** Responsible for the automated, programmatic submission of these manuscripts to conference management platforms.

This division of labor allows for modular development and enables the system to generate and submit papers with minimal human oversight, achieving the scale requirements for effective denominator inflation. We show the illustration of the threat model in Figure 4.

### 3.1. The Research Agent for Plausible Paper Generation

The Research Agent leverages state-of-the-art Large Language Models (LLMs) to generate papers that are **not scientifically valid but are stylistically plausible**. The success of the threat hinges on this crucial distinction. There have been many existing works on Research Agents, demonstrating end-to-end pipelines that can autonomously generate research ideas, design and run experiments, analyze results, and draft full papers (Ren et al., 2025; Huang et al., 2025; Lee et al., 2024). The Research Agent in the Agentic Conference Denominator Gaming can be implemented based on them. The core is to adapt the generated papers to **bypass desk-rejection scrutiny**. Specifically, the agent need to first download the LaTeX template from the targeted conference's website, and then meticulously analyze the required style from the *.sty* and *.tex* files. This allows it to create a perfect, camouflaged paper skeleton. By acquiring this crucial blueprint, the agent effectively crams its content into the correct structure, making the final document superficially indistinguishable from genuine research. This process raises the bar for initial rejection based on formatting or structural flaws, forcing busy human reviewers to evaluate documents that appear legitimate at first glance.

### 3.2. The Submission Agent for Automated OpenReview Flooding

The Submission Agent automates the logistical and often tedious process of submitting a paper to a modern conference management platform. The widespread adoption of platforms like OpenReview, which provide comprehensive and well-documented Python APIs, makes this component

highly feasible (Kim et al., 2025). These APIs, created to enhance efficiency for legitimate researchers, become a double-edged sword. The very infrastructure built to manage the submission growth is also the infrastructure that enables a scaled, automated attack.

Next, we decompose the process and show the technical points of the submission agent.

- **Account Registration:** The agent is programmed to automate registration with a fixed web interface and connect to a designated mailbox for verification. Educational emails can expedite this process. Through preliminary investigation, we find that some educational mails can be obtained or generated massively, enabling Sybil attacks.

- **Conference Form Completion:** The agent can automatically extract and populate submission details of the generated paper based on the OpenReview Submission Form[1], which is structured in JSON. Moreover, the agent can adapt to the variable forms and specific demands of different conferences efficiently. Specifically, when a conference is targeted, the agent performs detailed web collection, gathering and customizing all necessary submission data (*e.g.* abstract and author info). The entire submission sequence is ultimately executed using the OpenReview-Client Python library[2].

- **Rebuttal Stage:** When the Rebuttal Stage is available, the agent can fetch the reviews and respond to them directly via OpenReview APIs. As the attack does not aim for acceptance, careful rebuttals are unnecessary, or can achieved by existing scientific agent for rebuttal (Ma et al., 2026).

*Practical feasibility validation.* To verify that the proposed pipeline is not merely conceptual, we conduct a tightly controlled end-to-end feasibility test. Specifically, we utilize an existing scientific agent to complete a full paper draft, and implement a submission agent to register on OpenReview and automatically submit the manuscript through the OpenReview submission workflow. This experiment confirms the executability of the *generation–registration–submission* chain in practice, achieved by the multi-agent system. Following responsible disclosure principles, the test was performed only once and at minimal scale, and the submission was promptly removed by the authors to avoid any lasting burden or disruption to the platform.

[1] https://docs.openreview.net/reference/api-v2

[2] https://github.com/openreview/openreview-py

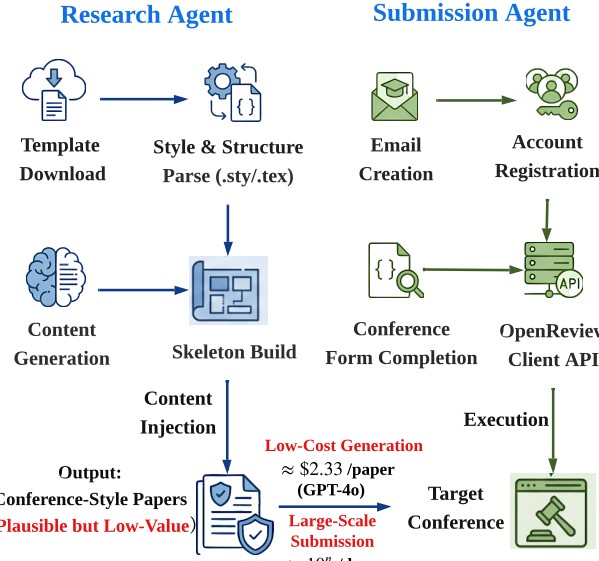

*Figure 4.* A proof-of-concept threat model for Agentic Conference Denominator Gaming. It can be achieved by a multi-agent pipeline, consisting of two agents: *research agent* for plausible-but-low-value paper generation and *submission agent* for automated OpenReview Flooding.

## 4. Broader Impacts and Systemic Risks

The successful execution of Agentic Conference Denominator Gaming would have profound and damaging consequences that extend far beyond the immediate outcome of a single conference. It represents a systemic threat to the integrity, sustainability, and trustworthiness of the academic publication process.

### 4.1. The Asymmetric Economics of Attack and Defense

This attack is characterized by a severe economic asymmetry that heavily favors the adversary. The attacker's primary burden is merely the financial expense of commercial LLM APIs used for paper generation. This cost is quantifiable and, as shown by (Schmidgall et al.; Schmidgall & Moor; Lu et al., 2024), remarkably low. Specifically, GPT-4o can complete the task in 1,165.4 s for merely $2.33, delivering a 94.3% success rate (Schmidgall et al.). Moreover, our preliminary analysis indicates that the submission agent costs less than one dollar and incurs negligible time overhead, since it mainly leverages the free OpenReview API.

In contrast, the cost of defense is borne by the academic community, manifesting as thousands of hours of high-skilled, uncompensated labor from volunteer reviewers and chairs (Paul Esau, 2025; Welsh, 2010). This asymmetry is compounded by the reusability of the synthetic corpus. Once generated, this manuscripts becomes a persistent digital asset that can be submitted to multiple conferences in sequence with near-zero marginal cost. This transforms

the threat from a single-event problem into a cross-venue, ecosystem-wide challenge, forcing each conference to erect its own defenses or risk becoming the weakest link. The economic disparity is stark: a minimal financial investment by a single actor can impose millions of dollars in opportunity costs on the collective academic enterprise.

## 4.2. Exacerbating Reviewer Burnout and Eroding Scientific Trust

The academic community is already operating at the limits of its cognitive capacity, grappling with a severe crisis of reviewer fatigue (Sharm, 2025). The pressure of the *publish or perish* culture has led to an explosion in submissions, placing an unsustainable burden on the limited pool of qualified reviewers (River, 2025; Hanson et al., 2024). Data suggests that approximately 20% of researchers handle over 90% of the peer-review workload, leading to widespread burnout (So, 2025). In this fragile ecosystem, Agentic Conference Denominator Gaming acts not merely as a disruption, but as a catastrophic accelerant to systemic collapse.

An influx of thousands of low-quality, AI-generated papers would dramatically lower the signal-to-noise ratio within the review pool. Reviewers, who are already asked to handle many papers per conference, would find themselves spending a significant portion of their time identifying and documenting flaws in papers that were never intended to be scientifically valid (So, 2025). This generation of *cognitive waste* fosters a profound sense of futility. The most experienced reviewers, whose time is most valuable, can be incentivized to withdraw from service. The resulting brain drain further diminishes the quality of feedback for legitimate authors, as the remaining pool becomes increasingly overwhelmed and less qualified (Sharm, 2025).

This degradation could lead to a severe decline in reputation. As the quality of the review process declines and more borderline papers are accepted due to the manipulated threshold by the threat, the overall quality of the conference proceedings would diminish. A conference's prestige is a direct function of the perceived quality of its published work. As this prestige erodes, top researchers would choose to submit their best work to other venues, further lowering the quality of submissions and accelerating the decline. This vicious cycle threatens to undermine the very trust that makes a publication at a top-tier conference a meaningful signal of scientific achievement.

## 4.3. From Paper Mills to Agent-driven Paper Mills: The Automation of Academic Fraud

Agentic Conference Denominator Gaming must be contextualized within the broader context of *industrialized academic fraud*. For years, publishers and integrity experts have battled *paper mills*, which are covert, profit-driven organizations that manufacture fraudulent manuscripts and sell authorship to researchers who need publications for career advancement (Bricker-Anthony & Herzog, 2023). These entities already pose a serious threat, systematically polluting the scientific record with fabricated data and plagiarized text (Byrne et al., 2024; Lin et al., 2025a).

The system we describe represents the next logical, and far more dangerous, step in this evolution: the transition from human-driven paper mills to fully automated **agent-driven paper mills**. We define an agent mill as a coordinated system of autonomous or semi-autonomous AI agents engineered to generate and submit fraudulent academic papers at unprecedented scale. Fueled by the generative capabilities of LLMs, this shift turns academic fraud from a labour-intensive cottage industry into an automated, assembly-line operation.

Specifically, the shift from paper mill to agent-driven paper mill transforms the problem of academic fraud in three fundamental ways: **scale**, **cost**, and **sophistication**.

- **Scale and Velocity:** Where human-driven paper mills produce hundreds or perhaps thousands of fraudulent papers per year, an agent-driven paper mill could generate that many in a single day. This creates a volumetric threat capable of overwhelming the entire academic publishing infrastructure, from submission portals to the finite pool of human peer reviewers.

- **Cost Collapse:** By replacing expensive human labor (writers, data manipulators) with comparatively cheap LLM API calls, the cost of producing a fraudulent paper plummets by orders of magnitude. This democratization of academic fraud makes the business accessible to a much wider range of malicious actors, from sole operators to state-sponsored entities, who can now operate at an industrial scale with minimal investment.

- **Sophistication and Evasion:** Agent-driven paper mills can systematically evade existing detection tools. By generating unique text, novel (though fabricated) data, and plausible figures for every paper, they neutralize plagiarism and template-based checks. Advanced agents could create webs of self-citing papers to invent legitimacy or even adapt their tactics based on reviewer feedback to become progressively harder to detect.

This escalation moves the threat from a manageable (though serious) problem of policing individual misconduct to a systemic crisis of epistemic security. The core challenge is no longer identifying a fake paper, but rather trusting that any paper is real when the cost of generating a plausible fake approaches zero. This threatens to create a recursive feedback loop where fraudulent, AI-generated papers are published,

indexed, and then used to train the next generation of LLMs, polluting the information ecosystem at its source.

# 5. Potential Defenses Strategies

Addressing the threat of Agentic Conference Denominator Gaming requires a multi-layered strategy that combines technical, policy-based, and systemic reforms. While recent author policies of many conferences have begun to incorporate preliminary safeguards, such as requirements of AI usage disclosures and stricter checklist enforcement (AAAI26CFP, 2025; ICML26CFP, 2025; NeurIPS25CFP, 2025), these measures may be insufficient against a scalable, adversarial actor committed to gaming the system. We propose several candidate defense strategies, and systematically evaluate their strengths and weaknesses. Our discussions remain open-ended and are aimed to offer actionable insights for the broader community.

## 5.1. Technical Defenses

As is illustrated in Section 3, Agentic Conference Denominator Gaming can be achieved by the collaboration of the scientific agent and the submission agent. We propose corresponding technical defenses to both of them:

- **Defense against malicious scientific agents.** Deploying AI-generated text detectors is the most immediate defense against the threat. Given that the peer-review system is already overwhelmed by submission growth, some form of automated triage is a strategic necessity to shield the limited pool of human reviewers from a flood of fraudulent papers. To reduce reliance on any single heuristic, conferences can combine detector scores with complementary, harder-to-game signals, *e.g.*, cross-submission similarity and template clustering (ICML26CFP, 2025), citation and artifact sanity checks (existence and consistency) (Paul Esau, 2025), and metadata-based anomaly detection (suspicious burst patterns, repeated authorship structures, or unusually correlated writing artefacts) (Lin et al., 2025a; Chandra et al., 2024).

- **Defense against malicious submission agents.** As is noted in recent paper (Lin et al., 2025b), agents have become the new entrance for digital traffic. Platforms such as OpenReview, should prevent automated large-scale account creation and submission. To this end, platforms can introduce stronger human-presence and identity assurance mechanisms, *e.g.*, requiring verified real-person authentication (a practical *Turing test*) during account registration and submission. Beyond basic CAPTCHAs, platforms could adopt layered friction such as ORCID verification, stronger rate limits and quotas per verified identity, reputation-based submission privileges for newly created accounts, and randomized manual audits for suspicious bursts. These measures aim to raise the marginal cost of scaling the submission agent, thereby mitigating denominator inflation at its source.

However, technical solution is fundamentally flawed and ultimately unsustainable when relied upon for high-stakes academic decisions (Kim et al., 2025). Current detection tools are notoriously unreliable, suffering from high false-positive rates that create an unacceptable risk of false accusations (Popkov & Barrett, 2025; Gotoman et al., 2025; Liang et al., 2023; Mathewson, 2023). The potential for catastrophic reputational and legal damage from a single false positive is so severe that many organizations have refused to use these tools to enforce their policies (Coley, 2023; Nelson, 2023).

Furthermore, relying on detection is a losing strategy in a perpetual cat-and-mouse game with generation technologies (Madaan, 2025). For every new detection method, a corresponding evasion technique is developed. The market is already saturated with commercial *AI humanizer* tools designed to bypass detectors. More critically, advanced adversarial attacks can fool even robust models. For example, *Adversarial Paraphrasing* (Cheng et al., 2025) uses a detector's own scoring mechanism to guide an LLM in rewriting text to be maximally human-like, reducing detection rates by over 98% and rendering the tool useless.

The current state of AI detectors is inherently flawed, characterized by fundamental unreliability, systemic biases, and a susceptibility to adversarial attacks. As such, a shift toward political and systemic measures is imperative.

## 5.2. Policy-Based Defenses

A more robust set of defenses involves altering the policies and rules of the submission pipeline to make the threat less feasible or effective.

- **Submission Fees:** Introducing a nominal submission fee is a controversial but potentially powerful experimental idea, which has adopted in IJCAI 2026 (IJCAI26CFP, 2025). A more equitable approach, inspired by conference registration policies, would be to require that at least one author per submission pays the fee. By imposing a financial cost, the conference inverts the economic asymmetry, rendering the high-volume flooding required for denominator gaming financially prohibitive. Moreover, this policy encourages higher-quality submissions from all authors. As submission is no longer a free lottery, authors are more motivated to ensure their work is polished and ready for review, rather than submitting rushed or speculative papers. This acts as a commitment mechanism that can

help control the total submission number and reduce reviewers' burden. As an additional benefit, the collected revenue from these fees could be used to serve as a reward for reviewers, providing a tangible incentive that could help mitigate the crisis of volunteer reviewer burnout (Kim et al., 2025). However, this defense has clear disadvantages. Legitimate human authors may object to paying for the labor of submission, and a nominal fee can create a barrier to entry for under-funded researchers, raising significant equity concerns.

- **Multi-Stage Review:** Adopting a refined multi-stage review process acts as an effective triage system. In an initial stage, all papers undergo a full review focused on identifying and rejecting submissions that are fundamentally flawed, scientifically unsound, or nonsensical, the very category into which papers generated by the threat would fall. Crucially, rejecting these papers can prevent them from consuming further resources. A smaller, pre-vetted cohort of promising papers then advances to a second stage for in-depth discussion and the traditional author rebuttal. This model, inspired by policies at conferences like AAAI (AAAI26Review, 2025), concentrates the finite and precious resource of reviewer attention on the most deserving manuscripts, making the entire system more resilient to floods of low-quality content.

### 5.3. Systemic Defenses

The most effective defenses are those that address the root structural vulnerability exploited by the attack, *i.e.*, the relatively stable acceptance rate of conferences.

- **Acceptance Number Range:** A single yet powerful defense is to break the link between submission count and acceptances. Concretely, instead of implicitly tying acceptances to the size of the submission pool via a roughly stable acceptance rate, a conference could pre-commit to an explicit bounded number range, that is defined independently of submission volume. This directly neutralizes the attacker's premise: inflating the denominator no longer changes how many papers can be accepted, therefore cannot raise the acceptance probability of any targeted subset by flooding the system. However, this policy also introduces governance challenges. The core difficulty lies in determining and justifying the target range. In a year with an exceptionally strong pool of submissions, it could force the rejection of high-quality work, frustrating the community. The reform is politically fraught, as it requires a cultural shift away from the simple, albeit flawed, metric of the acceptance rate.

- **Reputation Systems:** A long-term systemic reform

is the implementation of a reputation system for authors, drawing inspiration from models like the arXiv endorsement system (arXiv; Jordi et al., 2016). In this model, an author's reputation score is built over time based on their history of high-quality submissions and reviews. New authors or those with a low reputation score would need an endorsement from a high-reputation community member to submit a paper. This creates a significant barrier against the automated, Sybil-like scaling required for denominator gaming. Furthermore, the system could incorporate penalties: if a paper is flagged and confirmed to be AI-generated or fraudulent, all listed authors would see a significant reduction in their reputation scores. This creates a strong disincentive for established researchers to attach their names to questionable work and makes it substantially more difficult for attackers to operate anonymously.

## 6. Alternative Views

A robust analysis of this threat requires engaging with potential counterarguments and alternative interpretations of the problem. This section addresses two key objections to our position.

### 6.1. AI-Generated Papers are Easily Detectable

This viewpoint posits that the threat is overblown because current or near-future AI detection technology will be sufficient to identify and filter fraudulent submissions automatically.

This argument fundamentally underestimates the limitations of AI detection and the dynamics of the adversarial arms race. As detailed in Section 5.1, existing detectors are not reliable enough for high-stakes academic screening, exhibiting biases and unacceptable false-positive rates. More importantly, the relationship between generation and detection is inherently asymmetric. For any given detection model that learns to identify statistical patterns in AI-generated text, a generator can be fine-tuned or prompted to avoid those specific patterns. This creates a reactive loop where defenders are always one step behind attackers. Purely relying on a fragile technical fix for a systemic problem is an imprudent and ultimately losing strategy.

### 6.2. We Can Simply Defense by Implementing a Pre-Screening Phase to Filter AI Papers

A common pragmatic counter-proposal is to institute a separate triage or pre-screening phase. The goal is to identify and desk-reject suspected AI-generated submissions before they enter the formal review pool, thereby preventing them from inflating the submission denominator.

While intuitively appealing, this approach is flawed for three

critical reasons: **(1) Displacement of Burden:** This strategy does not eliminate the resource drain, it merely shifts the bottleneck to the Chairs who has the power to dest-reject submissions. Manually triaging thousands of flagged submissions requires immense cognitive labor. This diverts leadership from critical decision-making to administrative policing, effectively paralyzing the conference's executive capacity even without a formal review process. **(2) Procedural Injustice and Bias:** Triage relies on rapid heuristics that inherently disadvantage legitimate authors, particularly non-native English speakers (Liang et al., 2023). Many researchers use AI tools for translation and stylistic polishing, which is a practice distinct from AI generation. Conflating the two risks high false-positive rates. Summarily rejecting human work without a full review is not only discriminatory but deeply disrespectful to the authors' labor, potentially demoralizing the community and discouraging future participation. **(3) Adversarial Adaptation:** A static screening phase creates a learnable optimization target. Once the triage criteria become predictable, attackers can fine-tune their agents to bypass these checks. This turns the defense into a temporary hurdle rather than a permanent solution, as the agents will simply evolve to meet the minimum threshold required to enter the official pool.

## 7. Conclusion

The reliance of AI conferences on relatively stable acceptance rates, coupled with a strained volunteer peer-review system, has exposed a critical systemic vulnerability. This paper has introduced the Agentic Conference Denominator Gaming as a practical, economically asymmetric threat where automated agents can flood submission pools to manipulate acceptance outcomes. The consequences are severe: accelerating reviewer burnout, eroding scientific prestige, and industrializing academic fraud into agent-driven paper mills. We contend that reactive technical fixes like AI detection are insufficient against a determined adversary. The most robust defense needs a combination of political and systemic strategies. This paper serves as a call to action for the research community to proactively implement more resilient policies and safeguard the integrity of our core scientific validation mechanisms before this vulnerability is widely exploited.

## Acknowledgments

This paper is supported by National Natural Science Foundation of China (624B2096, 72595872, 72542012, 62322603).

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

# A. More Discussions

## A.1. Attack Incentives

One may question why the attacker bears the costs themselves while the benefits are broadly shared among all legitimate authors. Our responses to this concern are as follows:

*First*, it is possible that the benefits of such an attack may partially accrue to other authors. However, these benefits are neither uniform nor guaranteed, and the attacker can strategically limit collateral effects through:

- **Subfield Targeting:** The attacker can restrict generated submissions to specific tracks or niche topics, concentrating the threshold drop on their own subfield and minimizing any incidental benefits for unrelated papers.

- **Poisoning Reciprocal review:** The accounts for fake submissions can be weaponized to infiltrate the reviewer pool. Leveraging these synthetic identities, an attacker could deliberately influence reviews of competing human-authored papers, actively suppressing competitors' chances.

*Second*, given the technical feasibility above, we believe that attacker's rationality in allowing competitors to incidentally benefit is primarily a **social-psychological** issue: the attacker's decision is driven by perceived personal gain and strategic opportunity, rather than concern for collateral effects on others.

Actually, this pattern is **already observed across other domains** involving the manipulation of shared statistical metrics. Bearing a private cost to manipulate a system for personal gains, even if it inadvertently benefits free-riding competitors, is a well-documented phenomenon. Two representative examples are:

- **Ticket Market Price Manipulation** (FTC, 2021): Attackers bore technical and legal costs to deploy bots, fake accounts, and IP masking to bypass Ticketmaster limits, creating artificial scarcity and inflating secondary market prices. Uninvolved resellers occasionally benefited from these distortions, but attackers remained indifferent and only focused on their own direct massive profits.

- **Social Media Influence Manipulation** (FTC, 2019): Actors used bots and fake accounts to inflate followers, views, and likes, distorting shared metrics and bypassing algorithmic visibility thresholds for commercial gain. While other creators occasionally benefited from exposure spillover, attackers' actions were rationalized by their own direct payoffs, making collateral effects irrelevant to their incentive.

In these scenarios, the fact that manipulating a shared statistical metric inevitably generates incidental benefits for others is a *structural byproduct* of the system, *rather than a psychological deterrent to the attacker.*

**Finally**, as emphasized in our title and throughout the paper, we are not proving or asserting that such attacks will inevitably occur. Rather, our goal is to demonstrate their technical feasibility and damaging effects, highlighting the importance of early awareness and preventive measures within the academic community.

### A.2. Emerging Signs of The Attack

We provide empirical evidence that low-quality AI-generated content is increasingly infiltrating major conference submission pools, underscoring the imminent risks of denominator gaming attacks. Specifically, we evaluate submissions to NeurIPS and ICLR from recent years by scoring their abstracts with a widely-adopted AI text detection model, *i.e.*, ai-text-detector-v1.01[3]. Higher detection scores correlate with an increased probability of AI involvement.

Tables 2 presents the detection scores for both accepted and non-accepted (including rejected and withdrawn) papers across different years, with the Average Annual Growth Rate (AAGR) reported to highlight the trend over time. Our key observations are as follows:

- **Higher AI involvement:** For both accepted and non-accepted papers, we observe a clear upward trend in AI detection scores, suggesting that AI-generated content is becoming more prevalent in the submissions pool.

- **Absolute score disparity:** Non-accepted papers typically have higher average detection scores than accepted papers, suggesting that AI-generated content is more common in lower-quality submissions that fail to meet the acceptance bar.

- **Faster increase for non-accepted papers:** The AAGR for non-accepted papers is remarkably higher than for accepted papers. This suggests that the increase in AI-generation content is more pronounced in rejected or withdrawn papers, indicating the emerging risks of the denominator gaming attack.

**Rating analysis.** Moreover, we also analyze post-rebuttal average ratings for ICLR submissions, with data from official sources and Paper Copilot (Yang et al., 2025a). A slight decline is observed in the average ratings for accepted papers ($6.59 \rightarrow 6.44 \rightarrow 6.43$ from 2023 to 2025). Conversely, the average rating for rejected papers has declined more noticeably ($4.90 \rightarrow 4.94 \rightarrow 4.53$), resulting in an overall

---

[3]https://huggingface.co/desklib/ai-text-detector-v1.01

*Table 2.* AI-generated scores of NeurIPS and ICLR submissions by ai-text-detector-v1.01.

| Year | 2022 | 2023 | 2024 | 2025 | 2026 | AAGR |
|---|---|---|---|---|---|---|
| NeurIPS (Accept) | 0.1239 | 0.2236 | 0.2872 | 0.4359 | - | 53.6% |
| NeurIPS (Non-accept) | 0.1077 | 0.2455 | 0.3159 | 0.4812 | - | 69.7% |
| ICLR (Accept) | 0.1222 | 0.1071 | 0.2234 | 0.3753 | 0.3804 | 41.4% |
| ICLR (Non-accept) | 0.1124 | 0.1095 | 0.2820 | 0.4384 | 0.4412 | 52.8% |

rating drop of the full submission pool ($5.44 \rightarrow 5.41 \rightarrow 5.14$). This pattern indicates that the submission pool is expanding with a larger fraction of lower-quality papers.

### A.3. Educational Email Plausibility

Regarding the feasibility of mass-acquiring verifiable educational email accounts, we intentionally avoid detailing specific mass-email generation methods for ethical reasons, as these are sensitive and could encourage misuse. However, we would like to provide more related literature that support the feasibility of acquiring such accounts massively.

Shah et al. investigate the issue of identity theft in AI conference peer reviews, uncovering several instances where bad actors exploited university email aliases to create fraudulent reviewer profiles. Gao et al. provide an in-depth analysis of the underground ecosystem that supports automated account registration bots. They explore how malicious actors use bots to create and manage large volumes of accounts, leveraging system vulnerabilities in platforms that rely on email-based verification. Kipchirchir et al. evaluate the security weaknesses in University SMIS, particularly how these vulnerabilities could be exploited to gain unauthorized access to student data.

