# OpenReview forum: "Position: Academic Conferences are Potentially Facing Denominator Gaming Caused by Fully Automated Scientific Agents"
_ICML.cc/2026/Position_Paper_Track — ICML 2026 Position Paper Track regular_

### Official Review · Reviewer_gfT8 · 2026-03-09

**Significance:** 4
**Argument Clarity:** 3
**Rating:** 4
**Confidence:** 3

**Questions:**

Please refers to the Weakness section listed above.

**Alternative Views Section:**

Yes

**Compliance With Llm Reviewing Policy A Conservative:**

Affirmed.

**Discussion Potential:**

3

**Paper Summary:**

This paper describes the threats AI conferences face from denominator gaming. It first provides evidence that top AI conferences maintain relatively stable acceptance rates while facing an exponentially increasing volume of submissions. It then demonstrates the possibility of using AI agents to generate and submit papers at minimal economic cost, illustrating how AI could be used to game the system. The paper further discusses the potentially severe consequences of this phenomenon, such as reviewer burnout and diluted research quality. Finally, it proposes several possible solutions, such as introducing administrative fees per paper, and calls for the community to take this problem seriously.

**Position:**

Yes

**Position In Title:**

Yes

**Related Work:**

3

**Strengths And Weaknesses:**

**Strengths:**

1. The problem studied in this paper is timely, important, and increasingly relevant to the research community. The rapid growth in submissions to major AI conferences has become a painful issue for many researchers, yet there is still no widely accepted solution.

2. The paper takes a clear position on this issue, presenting evidence to highlight the potential existence of the problem, discussing its potential severe impacts, and outlining several possible solutions.

**Weakness:**

1. The motivation of individual participants is not clearly articulated. While submitting more papers may increase the overall number of submissions, it does not necessarily increase the acceptance probability of any individual paper. In other words, entering the “game” by submitting additional or low-quality papers may not directly benefit the attacker. Similarly, attacking the system by submitting many fake or AI-generated papers does not automatically increase the likelihood that the attacker’s own paper will be accepted. Therefore, the incentive structure behind such behavior remains unclear. To strengthen the argument, the paper should provide more concrete evidence that such scenarios have already occurred or are likely to occur in practice. In particular, the causal connection between attackers’ actions and the increased acceptance probability of their targeted papers needs to be clearly established. **What incentives do individual participants have to submit large numbers of fake or AI-generated papers, given that doing so does not necessarily increase the acceptance probability of their own papers? Can the authors clarify the mechanism by which flooding the submission pool with fake papers would increase the likelihood that a specific targeted paper is accepted?**

2. The increase in submission volume may not necessarily be driven by AI-generated papers. Instead, it could result from several other factors, such as “Fibonacci-style” submission strategies (e.g., submitting to multiple venues sequentially), the resubmission of previously rejected papers from other conferences, increased research productivity, and the growing attention that AI conferences receive from researchers in diverse fields, including social sciences and biomedical sciences. These factors could naturally lead to higher submission volumes without implying systemic attacks. Therefore, it would be helpful for the authors to provide evidence that the rising submission volume is indeed linked to AI-generated or adversarial submissions. In particular, **is there any empirical evidence showing a decline in the average quality of accepted papers, or evidence that coordinated submission attacks have already occurred or are emerging in current AI conference ecosystems?**

**Support:**

3

---

> ### Author Rebuttal · Authors · 2026-03-31
>
> Dear Reviewer gfT8,
>
> Thanks for your valuable feedback and thoughtful questions. We hope our responses below can answer your questions.
>
> ---
> ## W1: Attack incentive clarification
>
> We would like to clarify that the attack can still be individually rational because the benefit need not be global or uniform.
> - **Targeted attack in vulnerable subfields:** The attack can be strategically focused rather than indiscriminate. Attackers may concentrate AI-generated submissions in specific topics, tracks, or subfields where their own borderline paper is competing, thereby perturbing the *local* competitive environment relevant to their target manuscript. This effect is even more pronounced in *specialized or niche areas*, where the overall number of submissions is limited. In such settings, injecting additional low-quality papers can more easily shift the local acceptance threshold in the attacker’s favor. As a result, smaller fields are likely to experience greater disruption, allowing the attackers to internalize the benefits.
> - **Low cost and increasing attack feasibility:** As discussed in Sec. 4.1,  the attacker bears low API/generation cost, since the AI-generated papers are intentionally low-quality. With LLM generation  becoming more efficient and cheaper, it will be increasingly easy and cost-effective to conduct the attack. Moreover, once the synthetic corpuses are generated, they can be *reused* across multiple conferences with near-zero marginal cost, so even a small expected gain can become worthwhile over repeated submission cycles.
>
> Overall, we believe that both gaming and flooding are realistic risks rather than purely theoretical ones. As empirical statistics show below, there are already emerging signals indicating this trend. More importantly, if such behavior becomes widespread, its disruptive effect on the current peer-review system could be severe. Our position in this paper is precisely to point out this systemic risk early and to timely encourage community attention and discussion before it becomes routine.
>
> ---
> ## W2: Emerging signs of the attack
>
> We provide empirical evidence that low-quality AI-generated content is increasingly infiltrating major conference submission pools, underscoring the imminent risks of denominator gaming attacks.
> Specifically, we evaluate submissions to NeurIPS and ICLR from recent years by scoring their abstracts with a widely-adopted AI text detection model, i.e., ai-text-detector-v1.01. Higher detection scores correlate with an increased probability of AI involvement. **Tables 1** presents the detection scores for both accepted and non-accepted (including rejected and withdrawn) papers across different years, with the Average Annual Growth Rate (AAGR) reported to highlight the trend over time.
> > **Table 1: AI-generated scores by ai-text-detector-v1.01**
>
> | Year | 2022   | 2023   | 2024   | 2025   | 2026   | AAGR   |
> |-|-|-|-|-|-|-|
> | NeurIPS (Accept) | 0.1239 | 0.2236 | 0.2872 | 0.4359 | -      | 53.6%  |
> | NeurIPS (Non-accept) | 0.1077 | 0.2455 | 0.3159 | 0.4812 | -      | 69.7%  |
> | ICLR (Accept)    | 0.1222 | 0.1071 | 0.2234 | 0.3753 | 0.3804 | 41.4%  |
> | ICLR (Non-accept)    | 0.1124 | 0.1095 | 0.2820 | 0.4384 | 0.4412 | 52.8%  |
>
> Our key observations are as follows:
> - **Higher AI involvement:** For both accepted and non-accepted papers, we observe a clear upward trend in AI detection scores, suggesting that AI-generated content is becoming more prevalent in the submissions pool.
> - **Absolute score disparity:** Non-accepted papers typically have higher average detection scores than accepted papers, suggesting that AI-generated content is more common in lower-quality submissions that fail to meet the acceptance bar.
> - **Faster increase for non-accepted papers:** The AAGR for non-accepted papers is remarkably higher than for accepted papers. This suggests that the increase in AI-generation content is more pronounced in rejected or withdrawn papers, indicating the emerging risks of the denominator gaming attack.
>
> **Rating analysis.**  Moreover, we also analyze post-rebuttal average ratings for ICLR submissions, with data from official sources and Paper Copilot. A slight decline is observed in the average ratings for accepted papers (6.59 → 6.44 → 6.43 from 2023 to 2025). Conversely, the average rating for rejected papers has **declined more noticeably** (4.90 → 4.94 → 4.53), resulting in an overall rating drop of the full submission pool (5.44 → 5.41 → 5.14). This pattern indicates that the submission pool is expanding with a larger fraction of lower-quality papers.

---

> > ### Author Rebuttal · Reviewer_gfT8 · 2026-04-01
> >
> > # Consider the following mathematical model
> >
> > Let there be n = 10 players. Each player initially submits one high-quality paper, and all high-quality papers are equally likely to be accepted. The venue accepts a fixed fraction $\alpha = 0.1$ of all submitted papers.
> >
> > Suppose player $ A $ additionally submits $k = 10$ low-quality papers, each of which has acceptance probability $0$. Then:
> >
> > - Number of high-quality papers: $ n = 10 $
> > - Number of low-quality papers submitted by $ A $: $ k = 10$
> > - Total number of submissions:
> >   $
> >   N = n + k = 20
> >   $
> > - Total number of accepted papers:
> >  $
> >   \alpha N = 0.1 \cdot 20 = 2
> >   $
> >
> > Since only the $ n = 10 $ high-quality papers can be accepted, and they are symmetric, each high-quality paper is accepted with probability
> > $
> > p(k) = \frac{\alpha (n + k)}{n}.
> > $
> >
> > In this example,
> > $
> > p(10) = \frac{0.1 (10 + 10)}{10} = \frac{2}{10} = 0.2.
> > $
> >
> > So after flooding, every player’s good paper, including $A$’s, has acceptance probability
> > $
> > 0.2.
> > $
> >
> > If player $A$ submits $k$ worthless papers, then each good paper’s acceptance probability becomes
> > $
> > p(k) = \frac{\alpha (n + k)}{n},
> > $
> > as long as $ \alpha (n + k) \leq n $.
> >
> > **Question**:
> >
> > Why does player A have an incentive to take a privately costly action $k > 0$ that raises the acceptance probability
> > $p(k) = \frac{\alpha (n + k)}{n}$ **for everyone equally**, **including competitors**?
> >
> > That's the simplified scenario. If the competitor's paper quality is also on the borderline, and the quality is even slightly better, the rise of their paper's acceptance probability might be even higher than acceptance probability of player A who takes the costly action.
> >
> > **Please justify the rationality behind the player A, who bears the cost himself but benefits everyone, including his competitors.**

---

### Official Review · Reviewer_UGrF · 2026-03-12

**Significance:** 4
**Argument Clarity:** 2
**Rating:** 4
**Confidence:** 4

**Questions:**

My main issue with this paper is that I don't totally buy the denominator gaming premise as I describe above - any responses on this would be helpful.

**Alternative Views Section:**

Yes

**Compliance With Llm Reviewing Policy A Conservative:**

Affirmed.

**Discussion Potential:**

4

**Paper Summary:**

The paper suggests that agents will be used to "denominator game" academic conferences - inflating total numbers of submissions in order to increase acceptance probability, given a fixed acceptance rate policy. They authors describe how an agentic setup would work and argue the existing incentives are in place to make this likely. They discuss several potential defenses against this attack and some of the strengths and weaknesses with each.

**Position:**

Yes

**Position In Title:**

Yes

**Related Work:**

4

**Strengths And Weaknesses:**

I think this is a really important area of discussion and I appreciate the paper highlighting it. Some strengths:
- noting some of the incentives such as economic disparity between attacking and defense
- some good points about potential defenses, such as rate limiting, multi stage review, submission fees, and breaking the link from acceptances to submission count
- argues that the downsides to OpenReview having an API are significant (maybe it should be read only?)

A couple weaknesses I note in the argument:
- I'm not sure I buy the denominator gaming premise - it seems that the costs are borne by the attacker, while the benefits would accrue broadly to all legitimate submittors. This means that the incentives may not actually be very strong to do denominator gaming - the attacker will not benefit much, and all competitors in the academic marketplace will benefit equally
- "Through preliminary investigation, we find that some educational mails can be obtained or generated massively" - I'm not clear on what this means - don't you need access to a real education email inbox? Not sure I buy the mass generation of educational emails as an attack vector
- kind of a minor point but I don't think the phrase "agent mill" is right, should be "agent driven paper mill" or something (it's not a mill for agents)

**Support:**

2

---

> ### Author Rebuttal · Authors · 2026-03-31
>
> Dear Reviewer UGrF,
>
> Thanks very much for your supportive and insightful feedback. We hope our responses below can address your concerns.
>
> ---
> ## W1 & Q: Attack incentive clarification
> We would like to clarify that the attack can still be individually rational because the benefit need not be global or uniform.
> - **Targeted attack in vulnerable subfields:** The attack can be strategically focused rather than indiscriminate. Attackers may concentrate AI-generated submissions in specific topics, tracks, or subfields where their own borderline paper is competing, thereby perturbing the *local* competitive environment relevant to their target manuscript. This effect is even more pronounced in *specialized or niche areas*, where the overall number of submissions is limited. In such settings, injecting additional low-quality papers can more easily shift the local acceptance threshold in the attacker’s favor. As a result, smaller fields are likely to experience greater disruption, allowing the attackers to internalize the benefits.
> - **Low cost and increasing attack feasibility:** As discussed in Sec. 4.1,  the attacker bears low API/generation cost, since the AI-generated papers are intentionally low-quality. With LLM generation  becoming more efficient and cheaper, it will be increasingly easy and cost-effective to conduct the attack. Moreover, once the synthetic corpuses are generated, they can be *reused* across multiple conferences with near-zero marginal cost, so even a small expected gain can become worthwhile over repeated submission cycles.
>
> **Empirical study.**  We further evaluate recent NeurIPS and ICLR submissions by scoring their abstracts using a widely-adopted AI text detection model. We also analyze post-rebuttal average ratings for ICLR submissions. Due to the character limitation, please refer to our response to [Reviewer gfT8's Weakness 2](https://openreview.net/forum?id=36ITkZHUMZ&noteId=r7CwkkVFbt) for further details. Key observations are summarized as follows:
> - **Higher AI involvement:** Both accepted and non-accepted papers show an upward trend in AI detection scores, indicating increasing AI-generated content in submissions.
> - **Absolute score disparity:** Non-accepted papers have higher average detection scores than accepted papers, suggesting AI-generated content is more common in lower-quality submissions.
> - **Faster increase for non-accepted papers:** Non-accepted papers exhibit a higher average annual growth rate  in AI detection scores, highlighting a more pronounced rise in AI-generated content among rejected papers.
> - **Rating decline:** The average ratings for both accepted and rejected papers have declined over time, with a sharper drop in rejected papers, indicating an increasing presence of lower-quality submissions in the pool.
>
> Overall, we believe that both gaming and flooding are realistic risks rather than purely theoretical ones. Empirical studies above show that there are already emerging signals indicating this trend. More importantly, if such behavior becomes widespread, its disruptive effect on the current peer-review system could be severe. Our position in this paper is precisely to point out this systemic risk early and to encourage timely community attention and discussion before it becomes routine.
>
> ---
> ## W2: Educational email plausibility
> We thank the reviewer for the insightful concern regarding the feasibility of mass-acquiring verifiable educational email accounts. We intentionally avoid detailing specific mass-email generation methods for ethical reasons, as these are sensitive and could encourage misuse. However, we would like to provide more related literature that support the feasibility of acquiring such accounts massively, which will be included in the revision version of our paper.
>
> **[1] Identity Theft in AI Conference Peer Review.**
>
> Shah et al. investigate the issue of identity theft in AI conference peer reviews, uncovering several instances where bad actors exploited university email aliases to create fraudulent reviewer profiles.
>
> **[2] Demystifying the underground ecosystem of account registration bots.**
>
> Gao et al. provide an in-depth analysis of the underground ecosystem that supports automated account registration bots. They explore how malicious actors use bots to create and manage large volumes of accounts, leveraging system vulnerabilities in platforms that rely on email-based verification.
>
> **[3] Assessing Security Vulnerabilities in University Student Management Information Systems (SMIS) and Their Impact on Student Data Security.**
>
> Kipchirchir et al. evaluate the security weaknesses in University SMIS, particularly how these vulnerabilities could be exploited to gain unauthorized access to student data.
>
> ---
> ## W3: Phrase refinement
> We thank the reviewer for this helpful suggestion. We agree that the current phrasing can be improved, and we will revise it to “agent-driven paper mill” to make the phrase more precise and natural.

---

> > ### Author Rebuttal · Reviewer_UGrF · 2026-04-03
> >
> > Thanks for the responses. I think the notes clarifying attack incentive are pretty critical to the argument as presented by the paper - specifically, denominator gaming is really only rational in a fairly targeted sense where there are local thresholds - and this should be included in a significant way in the paper.
> >
> > The evidence that rejected papers with AI writing influence have increased is good and interesting in its own right although I don't think it constitutes evidence of denominator gaming - seems consistent to me with any world where cheap AI writing tools exist.
> >
> > I appreciate the desire to keep sensitive methods quiet - referencing relevant literature is I think a fine midway point to clarify the emails question.

---

### Official Review · Reviewer_Fx59 · 2026-03-14

**Significance:** 3
**Argument Clarity:** 3
**Rating:** 5
**Confidence:** 3

**Questions:**

1. How would the threat change if conferences move toward fixed acceptance numbers rather than rate-based targets?

2. Could this threat also push the community toward useful reforms (e.g. submission limits or reviewer compensation)?

**Alternative Views Section:**

Yes

**Compliance With Llm Reviewing Policy A Conservative:**

Affirmed.

**Discussion Potential:**

3

**Final Justification:**

The rebuttal addressed my concerns. I'll keep my score.

**Paper Summary:**

This paper introduces the idea of “Agentic Conference Denominator Gaming.” The main argument is that malicious actors could submit large numbers of AI-generated low-quality papers to inflate the denominator of submissions, which may increase the acceptance chance of a targeted set of legitimate papers if conferences keep relatively stable acceptance rates. The paper discusses feasibility of this threat and possible defenses.

**Position:**

Yes

**Position In Title:**

Yes

**Related Work:**

3

**Strengths And Weaknesses:**

***Strengths***

- The main idea is quite novel and interesting. Framing AI-paper flooding as a manipulation of the acceptance-rate mechanism (rather than only a quality issue) is a good insight.

-For a position paper, the authors provide some useful empirical grounding, e.g. acceptance rate stability across conferences and rough cost estimates for generating submissions.

-The discussion of defenses is fairly balanced, especially the point taht technical detection alone probably wont be enough.

***Weaknesses***

The paper would benefit from a more explicit quantitative analysis. Currently the argument about denominator inflation affecting acceptance probability is mostly intuitive and not really formalized.

The alternative views/counterarguments section feels a bit short given that some of the suggested policies (like submission fees) are somewhat controversial.

**Support:**

3

---

> ### Author Rebuttal · Authors · 2026-03-31
>
> Dear Reviewer Fx59,
>
> Thanks for your constructive feedback. We hope our responses below can address your questions.
>
> ---
> ## W1: Mathematical formalization
> We provide a simple mathematical formalization to help clarify how denominator inflation affects the acceptance rate.
>
> Let $N_h$ and $N_a$ denote the number of human and AI-generated submissions, respectively. The total number of submissions is $N_{sub} = N_h + N_a$.
>
> Suppose the conference aims to maintain a stable target acceptance rate, denoted by $\alpha$. The total number of accepted papers $N_{acc}$ is determined by:
> $$N_{acc} = \alpha \cdot (N_h + N_a)$$
> The total accepted set consists of accepted human papers $N_{acc}^h$ and accepted AI papers $N_{acc}^a$:
> $$N_{acc} = N_{acc}^h + N_{acc}^a$$
> Since the AI-generated papers are intentionally low-quality, we assume they are almost always rejected during the review process. Thus, $N_{acc}^a \approx 0$, and the quota for accepted papers is filled almost entirely by human submissions:
> $$N_{acc}^h \approx N_{acc} = \alpha \cdot (N_h + N_a)$$
> The effective acceptance rate for human submissions, denoted by $\hat{\alpha}$, then becomes:
> $$\hat{\alpha} = \frac{N_{acc}^h}{N_h} \approx \frac{\alpha \cdot (N_h + N_a)}{N_h} = \alpha \cdot \left( 1 + \frac{N_a}{N_h} \right)$$
> As the ratio of AI submissions to human submissions $N_a / N_h$ increases, **the effective human acceptance rate** $\hat{\alpha}$ rises proportionally, even if the actual quality of human papers remains unchanged.
>
> ---
> ## W2: Alternative view expansion
> We would like to clarify that our discussion of submission fees is intended as a defensive measure (as detailed in Sec. 5.2, where we weigh its pros and cons) rather than an alternative view, since it actually agrees with our core position and is motivated by the same underlying concern of our paper.
>
> To address your point, we will add the following perspectives to our revision:
>
> **Additional View 1: Immunity of High-Quality Papers.** One might argue that outstanding papers will be accepted regardless of total submissions, as they clearly stand out from low-quality AI submissions.
>
> **A:** The goal of the attack is not to displace top-tier works, but to lower the bar for borderline submissions. Moreover, as documented in recent works [1,2], by leveraging mass-produced email aliases to create fraudulent identities, attackers can manipulate the review process, meaning that even high-quality papers are not immune.
>
> **Additional View 2: Reputation Risk as a Deterrent.** One might argue that the risk of being caught gaming the system is too high for attackers, acting as a natural deterrent against such attacks.
>
> **A:** This underestimates the difficulty of attribution. Attackers can use agents to generate diverse writing styles and submit via decoupled identities, making attribution nearly impossible. Since there is no traceable link between the AI-generated papers and the attacker’s identity, the reputational cost is effectively zero.
>
> ---
> ## Q1:   Impact of fixed acceptance numbers on the threat
> If conferences move toward fixed acceptance numbers, the specific attack mechanism would be substantially weakened. Attackers could no longer rely on inflating submission volumes to shift the acceptance threshold through a stable submission rate.
>
> However, the broader threat would not disappear entirely. Even with fixed acceptance counts, attackers could still **manipulate the review process**.  Existing works [1, 2] show that malicious actors already exploit submission systems through fraudulent identities and email aliases to influence reviews.  Even with fixed acceptance numbers, attackers could still disproportionately affect the decision-making process.
>
> Moreover, the rise of fully automated AI scientists could exacerbate this issue, as AI systems generating large volumes of papers may further enable manipulation, with fewer human reviewers able to assess all submissions effectively. We will discuss this further in the revision of our paper.
>
> ---
> ## Q2: Potential reforms prompted by the threat
> Yes, we believe that articulating this threat model can indeed drive the community toward useful reforms that could strengthen the conference ecosystem.
> As discussed in Sec. 5, we propose several reforms that could help improve the robustness of the review process. Some of these measures are already emerging in practice, e.g. submission fees, submission limits, and reputation system. For example, IJCAI 2026 has introduced a submission-fee policy, charging a USD 100 fee for each submission (waived for primary papers). This is explicitly framed as a way to support the reviewing community.
>
> In the future, we anticipate that more comprehensive defensive measures will emerge to address these challenges, and our paper aims to spark these discussions before the attacks become widespread.
>
> ---
> [1] Identity Theft in AI Conference Peer Review
>
> [2] Demystifying the underground ecosystem of account registration bots

---

> > ### Author Rebuttal · Reviewer_Fx59 · 2026-04-06
> >
> > Thank you for the detailed response. The clarifications provided address my concerns well. I will keep my current rating.

---

### Decision · Program_Chairs · 2026-04-30

**Decision:**

Accept (regular)

**Comment:**

The paper presents a game-theoretic analysis of the submission of papers using an LLM, and show that even if all such papers are rejected, there is still an advantage to submitting them under the assumption that conferences keep their rejection rates similar.

All reviewers were positive about the idea, but cautious about accepting the axioms underlying the model. In particular, such attacks require a large and costly effort by a determined attacker, while the reward is general and benefits anyone who submits.

However, putting this idea out there that there is a benefit to submitting bad LLM papers to conferences, will increase the likelihood that bad actors throw in the occasional LLM paper in a non-systematic way.  -- In much the same way, the meme on /r/machinelearning about reviewers being overly harsh on papers to increase the chance of their own paper being accepted, actually increases the chance of reviewers being overly harsh.

Regardless of why people submit LLM-generated papers (and many will submit because they think the paper might be accepted), the description of the harms and need for adaptive rejection rates is relevant and the paper will likely encourage debate.